# Targeting Metabolic Vulnerabilities to Overcome Prostate Cancer Resistance: Dual Therapy with Apalutamide and Complex I Inhibition

**DOI:** 10.3390/cancers15235612

**Published:** 2023-11-28

**Authors:** Valentin Baumgartner, Dominik Schaer, Daniel Eberli, Souzan Salemi

**Affiliations:** 1Laboratory for Urologic Oncology and Stem Cell Therapy, Department of Urology, University Hospital Zurich, Wagistrasse 21, 8952 Schlieren, Switzerland; valentin.baumgartner@usz.ch (V.B.); daniel.eberli@usz.ch (D.E.); 2Division of Internal Medicine, University Hospital Zurich, Wagistrasse 12, 8952 Schlieren, Switzerland

**Keywords:** prostate cancer, metabolism, mitochondria, apalutamide, IACS-010759

## Abstract

**Simple Summary:**

Prostate cancer often becomes resistant to drug treatment, causing it to advance aggressively. This study aims to find new ways to treat this challenging cancer. We looked at different types of prostate cancer cells to understand their unique metabolic behaviors. Moreover, we tested a combination of two drugs, Apalutamide (ARN) and mitochondrial complex I inhibitor IACS-010759 (IACS), to see if it could slow down cancer cell growth. This research uncovers how prostate cancer cells use energy differently and suggests a promising approach to treat them by targeting their unique metabolism. This could be a big step forward in improving the treatment of advanced prostate cancer.

**Abstract:**

Prostate cancer (PCa) often becomes drug-treatment-resistant, posing a significant challenge to effective management. Although initial treatment with androgen deprivation therapy can control advanced PCa, subsequent resistance mechanisms allow tumor cells to continue growing, necessitating alternative approaches. This study delves into the specific metabolic dependencies of different PCa subtypes and explores the potential synergistic effects of combining androgen receptor (AR) inhibition (ARN with mitochondrial complex I inhibition (IACS)). We examined the metabolic behaviors of normal prostate epithelial cells (PNT1A), androgen-sensitive cells (LNCaP and C4-2), and androgen-independent cells (PC-3) when treated with ARN, IACS, or a combination. The results uncovered distinct mitochondrial activities across PCa subtypes, with androgen-dependent cells exhibiting heightened oxidative phosphorylation (OXPHOS). The combination of ARN and IACS significantly curbed cell proliferation in multiple PCa cell lines. Cellular bioenergetics analysis revealed that IACS reduced OXPHOS, while ARN hindered glycolysis in certain PCa cells. Additionally, galactose supplementation disrupted compensatory glycolytic mechanisms induced by metabolic reprogramming. Notably, glucose-deprived conditions heightened the sensitivity of PCa cells to mitochondrial inhibition, especially in the resistant PC-3 cells. Overall, this study illuminates the intricate interplay between AR signaling, metabolic adaptations, and treatment resistance in PCa. The findings offer valuable insights into subtype-specific metabolic profiles and propose a promising strategy to target PCa cells by exploiting their metabolic vulnerabilities.

## 1. Introduction

Prostate cancer (PCa) is currently the most diagnosed cancer and a leading cause of cancer-related morbidity and mortality among men [1,2]. Patients diagnosed with advanced PCa initially benefit from treatment with androgen deprivation therapy. However, after a primary tumor response, nearly all tumors progress to a castration-resistant state with subsequent emergence of metastasis and rising cancer fatality. These treatment-induced resistance mechanisms allow the malignant cells to sustain growth during disease progression. 

As a major hallmark of cancer, metabolic reprogramming allows the cancer cells to better adapt to harsh drug treatments, ensuring survival [3,4]. The androgen receptor (AR) is the master regulator of many energy-producing pathways in normal cells, and aberrant AR signaling drives tumorigenesis in PCa [5]. Despite the effort to inhibit the AR in advanced hormone-independent stages with apalutamide, darolutamide, or enzalutamide, some PCa cells are able to bypass these interventions through lineage plasticity [6] and AR mutations [7], leading to tumors that are more aggressive. Therefore, AR-targeted treatment approaches provoke metabolic adaptations in tumor cells to promote alternative anabolic pathways. This switch towards a different metabolic phenotype could be an important target to exploit specific metabolic dependencies and ultimately prevent the development of drug resistance.

In many cancer cells, the “Warburg effect” is a phenomenon that leads to the preferred utilization of glucose for energy production even if sufficient oxygen is available (aerobic glycolysis) [8]. Paradoxically, normal prostate epithelial cells represent best Warburg’s notion of aerobic glycolysis. Instead of relying on mitochondrial oxidative phosphorylation, they primarily depend on glycolysis as their fuel source. These unique metabolic properties relate perfectly to the prostate’s physiological function—the production and secretion of the prostatic fluid, which nourishes the sperm cells. Citrate is a vital component of the prostatic fluid, essential for maintaining pH. In contrast to other cells, normal prostate epithelial cells display a truncated tricarboxylic acid (TCA) cycle that allows the accumulation of citrate and the subsequent transport into the acinar lumen. High levels of zinc are responsible for the inhibition of the mitochondrial aconitase [9], resulting in an incomplete interconversion from citrate to isocitrate and the build-up of citrate levels. In early PCa development, normal citrate-producing cells switch towards citrate-oxidizing cells [10]. However, the metabolic dependency of different PCa subtypes and potential metabolic rewiring mechanisms upon antiandrogen treatments is not fully understood. Moreover, the re/activation of a TCA cycle in PCa might create vulnerabilities that could be exploited by targeting mitochondrial metabolism. 

In this study, we investigated PCa subtype-specific metabolic dependencies and the potential synergistic effects of apalutamide (ARN-509) together with IACS, a potent complex I inhibitor [11] in androgen-dependent and -independent PCa cell lines.

## 2. Materials and Methods

### 2.1. Pharmacological Compounds

Apalutamide (ARN) and IACS were both obtained from Selleck Chemicals (Houston, TX, USA). FCCP, oligomycin, and rotenone/antimycin A were all purchased from Agilent. Galactose was acquired from Sigma Aldrich (Buchs, Switzerland).

### 2.2. Cell Culture

PCa cell lines PNT1A, LNCaP, C4-2, and PC-3 were all purchased from LGC Standards GmbH (Wesel, Germany). C4-2 cells were a generous gift from Prof. Dr. med. George N. Thalmann (University Hospital Bern, Bern, Switzerland). Cells were cultivated in RPMI (Thermo Fisher Scientific, Waltham, MA, USA), supplemented with 10% fetal bovine serum (FBS, Sigma-Aldrich, St. Louis, MO, USA) and 1% Penicillin/Streptomycin (P/S, Gibco), and incubated at 37 °C with 5% CO_2_.

### 2.3. Galactose Medium Preparation

RPMI without glucose (Thermo Fisher Scientific) was complemented with the following additives: 10% FBS, 1% P/S, and 10 mM galactose. Cells were cultured in galactose medium for 24 h before starting the experiments.

### 2.4. Cell Proliferation Assay

The proliferative capacity was measured by CellTiter-Glo 2.0 assay (Promega, Madison, WI, USA). Cells were seeded in 96-well plates (TPP) at a density of 3000 cells/well and cultured for one day. The next day, cells were treated with ARN and/or IACS for 3 days. Briefly, a mix of CellTiter-Glo and culture medium (100 µL, 1:2) was added to the cells which were then lysed on an orbital shaker for 2 min. A quantity of 50 µL was then transferred to white 96-well plates (Cellstar), and luminescence was recorded by a Cytation 5 imaging reader (BioTek, Winooski, VT, USA).

### 2.5. Mitochondrial Substrate Utilization

The cell-type-specific preference for mitochondrial substrate metabolism was assessed by Mitoplate S-1 readout (Biolog). Precoated test substrates were dissolved in 30 µL of assay mix per well containing 2× Biolog MAS, 6× Redox Dye MC, 24× saponin (1200 µg/mL), sterile H2O and incubated for 1 h at 37 °C. The cell suspension was resuspended in 1× Biolog MAS. Cells were counted and seeded at a cell density of 3 × 10^4^ cells per well. The microplate was then loaded into a Cytation5 microplate reader for kinetic measurement of the redox dye reduction. OD at 590 nm and 750 nm were measured simultaneously in 5 min read intervals for a total of 4 h. At the end of the readout, OD750 was subtracted from OD590 for background correction, and the values were normalized to the no-substrate-containing wells. For statistical analysis, the mean slope after 2 h was used.

### 2.6. Extracellular Flux Analysis by Seahorse Assay

To study the impact of pharmacological compounds on the metabolic potential of prostate cancer cells, we treated PNT1A, LNCaP, C4-2, and PC-3 cells with ARN, IACS and assessed changes in mitochondrial oxygen consumption (OCR) and extracellular acidification (ECAR). Treated cells were subjected to a mitochondrial stress test (Mito Stress Kit, Agilent) by a Seahorse XFe24 extracellular flux analyzer (Agilent) according to the manufacturer’s instructions. Briefly, cells were seeded into XFe24 microplates (2 × 10^4^ cells/well) and incubated at 37 °C, 5% CO_2_. The following day, cells were treated with 25 µM of ARN, 10 nM of IACS, or combination and incubated for 24 h. The sensor cartridge of the utility plate was hydrated with H2O and incubated in a non-CO2 incubator overnight at 37 °C. The next day, H_2_O was exchanged to XF calibrant. The injection ports were loaded with mitochondrial stressors: Oligomycin (1.5 µM), carbonyl cyanide 4-(trifluoromethoxy) phenylhydrazone (FCCP; 1.5 µM for PNT1A, 1 µM for LNCaP, 1 µM for C4-2, and 0.5 µM for PC-3), and rotenone plus antimycin A (1 µM). Cells were washed twice with XF RPMI medium, supplemented with glucose (10 mM), pyruvate (1 mM), and L-glutamine (2 mM). Next, cells were placed in XF RPMI medium and incubated at 37 °C without CO_2_ for degassing. After calibration of the sensor cartridge, the utility place was replaced by the microplate containing the cells, and OCR and ECAR were measured in parallel by a Seahorse XFe24 analyzer. Upon completion, the cells of each well were stained with 1 µg/mL Hoechst 33342 (Thermo Fisher Scientific). Fluorescently labelled nuclei were captured using a Leica DMi8 microscope as whole-well stitched images. ImageJ software (1.54 f NIH, Maryland, USA) was used to quantify the nuclei, which were subsequently used to normalize OCR and ECAR parameters.

### 2.7. Monitoring of Glucose and Lactate Levels

Extracellular glucose and lactate levels were measured by the bioanalyzer Vi-CELL MetaFlex (Beckman Coulter, Pasadena, CA, USA). Cells were seeded in 24-well microplates (TPP) with 4 × 10^4^ cells/well. After 24 h, cells were treated for 3 days with 25 µM of ARN, 10 nM of IACS or combination. Then, 24 h after treatment start, 100 µL of cell culture medium was aspirated per well and immediately analyzed for glucose and lactate levels for three consecutive days. Normal RPMI medium without cells served as a quality control. 

### 2.8. Protein Analysis

Automated Western blotting (WES; ProteinSimple, San Jose, CA, USA) was used to measure relative protein expression levels. Protein lysates were prepared and examined in accordance with the manufacturer’s instructions. The antibodies used included rabbit anti-androgen receptor (1:100, cell signaling), mouse anti-NDUFB8 (NADH:ubiquinone oxidoreductase subunit B8, 1:50, Novus), and mouse anti-GAPDH (1:100, Novus), which served as a loading control. The immuno-detected proteins were quantitatively analyzed based on their molecular weight (MW) and the intensity of the chemiluminescent signal (area). Compass software (6.1.0) from ProteinSimple was employed to perform size- and charge-based analysis for the proteins of interest. To standardize relative protein expression, each sample was normalized to GAPDH.

### 2.9. Immunostaining of Mitochondrial Network

Cells were fixed with 4% Paraformaldehyde (PFA; Artechemis, Zofingen, Switzerland), subsequently washed with PBS, and incubated with 0.5% Triton X-100 (Sigma-Aldrich) in PBS at RT to permeabilize. After another washing step with PBS, 3% Bovine Serum Albumin (BSA; Sigma-Aldrich) with 1% Normal Goat Serum (NGS, Sigma-Aldrich) was added to the cells for 1 h at RT in order to block unspecific bindings of antibodies. Further, cells were incubated overnight at 4 °C with the primary antibody rabbit TOM20 (Novus, 1:100). The day after, secondary antibody plus DAPI (1:200, Sigma-Aldrich) were added to the cells after washing with PBS and then incubated for 1 h at RT in a humidified chamber in the dark. As immuno-labelled secondary antibody we used CY3 Anti-rabbit IgG (Sigma-Aldrich, 1:500). Then, the cells were washed in PBS, and mounting medium DAKO (Agilent, Santa Clara, CA, USA) was added to seal the slides with cover slips. Fluorescence was observed by a Leica DMi8 microscope, and fluorescence intensity was assessed in ImageJ.

### 2.10. Statistical Analysis

Results were analyzed by one-way ANOVA with Bonferroni’s post correction using GraphPad Prism (GraphPad Software, Inc., La Jolla, CA, USA, version 9.5.1). *p*-values < 0.05 were considered statistically significant. Drug synergy was determined by employing the Bliss independence model within the SynergyFinder 3.0 WebApp [12]. All data presented are expressed as means with corresponding standard error of the mean (±SEM).

## 3. Results

### 3.1. Subtype-Specific Mitochondrial Activity in Prostate Cancer Cell Lines

Differences in mitochondrial function and consumption of metabolic substrates in PCa cell lines were assessed by using the Mitoplate assay. Substrate utilization was tested in normal (PNT1A), androgen-dependent (LNCaP), and -independent (PC-3) cell lines. Purple color formation of the tetrazolium redox dye reduction indicated differences in preferential metabolism of 31 substrates involved in energy production between different PCa cell lines (Figure 1A). Kinetic analysis revealed that androgen-dependent LNCaP cells have an elevated metabolic consumption rate of TCA intermediates compared to PNT1A and PC-3 cells (Figure 1B). Quantification of the mean slope after 2 h showed significant differences in NADH- and FADH2-producing substrates (Figure 1C). Compared to PNT1A, LNCaP cells had significantly increased metabolism of succinate, fumarate, and malate (*p* < 0.0001). Furthermore, LNCaP cells expressed significantly higher NDUFB8 (subunit of complex I) protein levels compared to PNT1A and PC-3 cells. The higher rates of mitochondrial oxidation of TCA intermediates and NDUFB8 protein levels in LNCaP cells suggest a specific fingerprint towards oxidative phosphorylation (Figure 1D and Appendix A).

### 3.2. Synergistic Antiproliferative Effect of Androgen Receptor and Mitochondrial Complex I Inhibition

To exploit certain metabolic dependencies of PCa cells, we examined the synergistic effect of ARN and IACS on cell proliferation (Appendix A). IACS is a clinical-grade, potent complex I inhibitor, and ARN is an antiandrogen drug. To evaluate the proliferative capacity of different PCa cell lines upon drug treatment, we first tested various times and dosages. Drug treatments with 25 µM of ARN and 10 nM of IACS were selected as concentrations close to their growth inhibition at 50% (GI50) values for a treatment period of three days (Figure 2A–D). Treatment with ARN significantly decreased viability in androgen-dependent LNCaP and -sensitive C4-2 cells. ARN had no significant impact on viability in androgen-receptor-negative PC-3 cells. Importantly, cells treated with ARN plus IACS showed significantly reduced cell proliferation compared to ARN only (all cell lines) or IACS only (LNCaP, PC-3) and DMSO controls (all cell lines) (Figure 2A). In addition, lower IACS concentrations demonstrated increased synergy with ARN in contrast to the higher dose of 10 nM, where the potential cytotoxic effects of IACS became more pronounced (Figure 2D and Appendix A).

### 3.3. Cellular Bioenergetics Modulation via Pharmaceutical Inhibition of AR and OXPHOS

To shed more light on distinct metabolic properties upon treatment with ARN and/or IACS, we analyzed changes in oxygen consumption rates (OCR) and extracellular acidification rates (ECAR) between different PCa cell lines after 24 h treatment (Figure 3A). OCR is a measure for OXPHOS capacity, and ECAR relates to glycolysis. In all cell lines, IACS alone and in combination with ARN significantly decreased OCR (Figure 3B). Moreover, ARN significantly reduced the maximum respiratory only in LNCaP cells. Additionally, immunostaining for mitochondrial import receptor TOM20 revealed significantly elevated expression levels in LNCaP, C4-2, and PC-3 cells compared to PNT1A (Figure 3C,D).

Conversely, treatment with IACS and combination with ARN elevated ECAR levels in all PCa cell lines, indicating a compensation mechanism towards an increase in glycolysis (Figure 4A). Interestingly, ARN alone resulted in a marginal reduction in glycolysis in LNCaP and C4-2 cells; however, this difference did not reach statistical significance. To gain a more comprehensive understanding of these results, we measured extracellular glucose and lactate levels upon treatment. In line with the extracellular flux analysis, compensatory glycolysis was observed when cells were treated with IACS and a combination treatment (Figure 4B). Furthermore, the slight reduction in glycolysis that was observed in ECAR levels became more pronounced during the analysis of the medium following ARN treatment. Compared to the DMSO control, both LNCaP and C4-2 fermented significantly less lactate (*p* < 0.0001) after three days of treatment with ARN. To further investigate the metabolic properties of different PCa cell lines, we tested the sensitivity of PCa cells towards 2-deoxy-d-glucose (2-DG), a glucose analogue that inhibits glycolysis. Compared to the control, the addition of 1 mM of 2-DG decreased viability to 27.89% ± 3.02 in PNT1A, 47.98% ± 1.80 in LNCaP, 37.75% ± 0.68 in C4-2, and 40.42% ± 0.89 in PC-3 cells (Figure 4C).

### 3.4. Galactose Supplementation Reveals Metabolic Vulnerabilities of PCa

Since our results suggested that complex I inhibition rewires cellular metabolic pathways to rely more on glycolysis as an alternative form for energy production, we wondered whether the removal of glucose would have an additional impact on the proliferative capacity. Therefore, we treated cells with ARN and/or IACS in glucose-free medium, supplemented with galactose (GAL). Treatment with ARN and IACS in glucose-deprived conditions (GAL) decreased cell proliferation in PCa cell lines compared to glucose-supplemented medium (control) (Figure 5A). Strikingly, PC-3 cells which exhibited resistance to complex I inhibition at the tested IACS concentrations ranging from 1 to 10 nM in glucose-supplemented medium showed markedly decreased proliferation when cultured and treated in glucose-depleted medium (GAL). Extracellular flux analysis revealed that glucose withdrawal indeed reduced glycolysis in all cell lines tested (Figure 5B). Moreover, PCa cell lines LNCaP, C4-2, and PC-3, but not PNT1A, experienced a shift towards increased OXPHOS.

## 4. Discussion

In prostate cancer, the modulation of the androgen receptor plays a pivotal role in driving the progression of the disease to a castration-resistant state [13,14]. Antagonism of AR has been shown to alter vital metabolic processes [15]. These therapy-induced metabolic changes ultimately give rise to cancer cells that are resistant to current therapy options and lead to an even more aggressive advancement of the disease. In this study, we have identified important subtype-specific metabolic profiles of PCa cells. Further, we have shown that targeting distinct metabolic properties significantly impedes PCa cell growth.

It is well known that dysregulated AR signaling is a main driver of oncogenic transformation in PCa. However, treatment resistance in advanced PCa hampers efforts to disrupt aberrant AR-regulated disease progression through antagonizing AR. By mitochondrial substrate and extracellular flux analysis, we have shown that AR dependencies are linked to a higher reliance on OXPHOS. These results are in line with previous studies that reported reliance on aerobic glycolysis in epithelial prostate cells [16] and increased mitochondrial ATP production in early-stage PCa [17]. They demonstrated that AR transcriptionally regulates the mitochondrial pyruvate carrier. Importantly, in LNCaP, C4-2, and PC-3 cells, the mitochondrial mass marker TOM20 was significantly increased compared to PNT1A. These results corroborate our findings of increased OXPHOS dependency in PCa cells and confirm previous research that showcased TOM20’s role in enhancing mitochondrial metabolism within cancer cells [18].

Malfunctions in mitochondria have been associated with the development of tumors and the aggressiveness of cancer [19]. AR has been shown to localize to mitochondria through a 36-amino-acid-long mitochondrial localization sequence (MLS) [20]. In other studies, androgen stimulation resulted in an increase in aerobic glycolysis [5,21], possibly through increasing citrate levels, leading to a truncated TCA cycle [5]. These studies support our results, which demonstrated reduced glucose consumption and decreased lactate production following ARN treatment in androgen-dependent PCa. It is hypothesized that AR–mitochondria interactions are also responsible for regulating OXPHOS by destabilizing mitochondrial respiratory complexes [20,22], highlighting aberrant mitochondrial metabolic rewiring processes in PCa.

IACS reduced OXPHOS in all tested PCa cell lines, whereas cell proliferation decreased in all cell lines except for PC-3 cells. PC-3 cells represent advanced and metastatic cells that rely simultaneously on glycolysis and OXPHOS for energy production. This metabolic flexibility could therefore suggest resistance mechanisms to IACS treatment. Moreover, PNT1A cells showed a marked decrease in their proliferation upon IACS treatment. This could be attributed to their glycolytic profile, limited ATP production, and overall quiescent metabolism of prostate epithelial cells [23]. Indeed, PNT1A cells presented a high 2-DG sensitivity and relatively modest reduction in OXPHOS capacity compared to transformed PCa cells. In LNCaP and C4-2 cells, the increases in ECAR upon combinational treatment compared to IACS alone were slightly different from glucose media analysis where a possible dampening effect of ARN was detected. Alternative pathways such as the pentose phosphate pathway, which has been shown to be modulated by AR signaling [24], can influence the extracellular acidification rate and may explain these discrepancies. The metabolic effects observed with apalutamide in our study may not only be seen by using apalutamide but also by using other AR blockers such as enzalutamide or by mimicking androgen deprivation therapy (ADT). However, the specific effects and metabolic responses can vary depending on the drug’s mechanism of action and cellular context.

A recent phase I clinical trial evaluated IACS in leukemia and other solid tumors [25]. Despite a partial RECIST response in one advanced PCa patient, the trial was discontinued due to substantial peripheral neuropathy observed in some patients. In a reverse-translational in vivo experiment, the authors showed that histone deacetylase 6 (HDAC6) inhibition could alleviate the drug-induced peripheral neuropathy. Moreover, HDAC6 has been shown to reduce glycolytic metabolism in breast cancer cells [26]. Therefore, a strategy to prevent adverse effects could be by lowering glucose levels simultaneously, for example, by a ketogenic diet, which could stimulate anti-inflammatory signaling cascades and has been associated with reduced neuropathy in preclinical models [27,28].

In vitro cell culture medium generally contains very high amounts of glucose, incomparable to physiological levels. These conditions might mask potential pharmaceutical effects of mitochondrial toxins in glucose-addicted cells (known as the Crabtree effect [29]). To circumvent confounding results, cells can be cultured in glucose-free, galactose-supplemented medium. In contrast to glucose, galactose as an alternative energy source does not yield any net ATP and maintains metabolic activity. Notably, this glucose-deprivation strategy has been shown to force cells relying on OXPHOS to meet their energy demands. Enhancing OXPHOS metabolism of cells grown in galactose is often applied to study potential mitochondrial toxins and mitochondrial impairments. We showed by extracellular flux analysis that this sugar replacement strategy triggered a metabolic rewiring in PCa cells by decreasing glycolysis and increasing OXPHOS simultaneously, whereas normal PNT1A cells decreased glycolysis without increasing OXPHOS. These observed shifts in metabolism confirm the metabolic flexibility in cancer cells shown by others in myeloid leukemia and pancreatic cancer cells [30,31]. Furthermore, GAL-cultured PCa cells were significantly more susceptible to mitochondrial-targeted therapies by complex I inhibition compared to their glucose-cultured counterparts. Importantly, PC-3 cells that were resistant to IACS in glucose-rich medium were highly sensitized to complex I inhibition in GAL culture conditions. In a previous study, PC-3 cells which showed metformin (complex I inhibitor) resistance switched towards OXPHOS upon docetaxel resistance and subsequently became highly susceptible to metformin treatment [32]. Our data suggest that complex I inhibition has only little efficacy in PC-3 cells, likely attributed to their high metabolic plasticity. Therefore, drug-induced glycolytic compensation could be circumvented by replacing glucose with galactose, ultimately enhancing the drug effect and sensitizing PCa cells significantly to mitochondrial-targeted therapies.

## 5. Conclusions

In summary, our results indicate that the concurrent use of anti-AR therapy and OXPHOS inhibition has the potential to restrict drug resistance and hinder the progression towards metastatic disease. Targeting metabolic dependencies revealed a potent effect of IACS in reducing OXPHOS activity across all tested PCa cell lines, with relatively moderate benefits from the addition of ARN. Furthermore, dual therapy synergistically inhibited cell growth in LNCaP cells. However, potential toxicity arising from the nonspecific activity of IACS might raise concerns. Further studies in an in vivo setting are needed to evaluate the safety and efficacy profile associated with the proposed combination therapy.

## Figures and Tables

**Figure 1 cancers-15-05612-f001:**
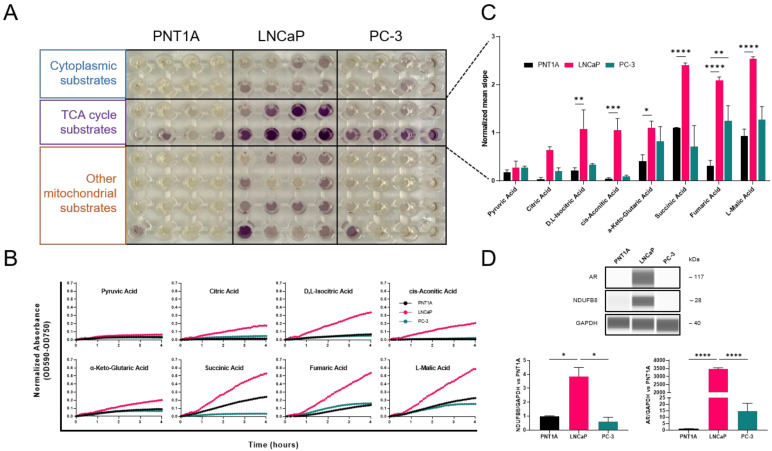
Mitochondrial activity in PCa. (**A**) Mitoplate assay. Representative image of plate layout after dye reduction (purple color formation) in PNT1A, LNCaP, and PC-3 cells. The intensity of color formation shows the utilization of a specific substrate in different energy-producing pathways. (**B**) Rate of dye reduction measured at OD590-OD750 and normalized to a no-substrate control for TCA cycle substrates over 4 h with 5 min measurement intervals. (**C**) Quantification of the mean slope after 2 h kinetic reading for different TCA cycle substrates. Values were normalized to a no-substrate control. Data from three independent experiments are presented as means ± SEM (*n* = 3). Significant differences relative to PNT1A are indicated by * *p* = 0.0289; ** *p* < 0.01; *** *p* < 0.001; **** *p* < 0.0001. (**D**) Representative virtual lanes of AR and NDUFB8 for PNT1A, LNCaP, and PC-3 (**top**). Quantification of relative protein expression levels normalized to GAPDH (*n* = 3) (**bottom**). The uncropped bolts are shown in Appendix A.

**Figure 2 cancers-15-05612-f002:**
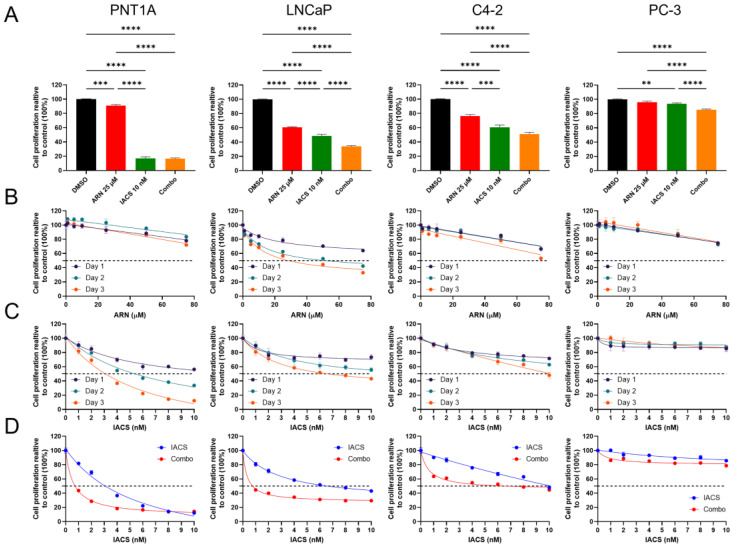
Treatment-induced changes in tumor cell growth. (**A**) Cell proliferation of PCa cells upon treatment with DMSO, 25 µM of ARN, 10 nM of IACS, and combo (ARN + IACS). Data are presented as means ± SD. Three independent experiments with three technical replicates each per condition are shown (*n* = 9). Statistical significance is shown by asterisks; ** *p* < 0.01; *** *p* < 0.001; **** *p* < 0.0001. Dose–response curve of ARN (**B**) and IACS (**C**) with indicated doses for a treatment period of 1, 2, and 3 days (*n* = 3). (**D**) Comparison in cell proliferation of IACS alone versus combination treatment at IACS concentration 1–10 nM (*n* = 3).

**Figure 3 cancers-15-05612-f003:**
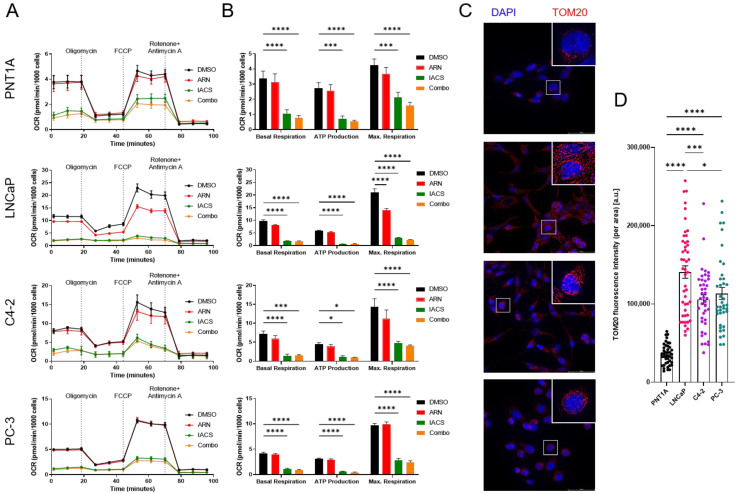
Alterations in cellular respiration upon AR and complex I inhibition. (**A**) Normalized oxygen consumption rates for different PCa cell lines after sequential addition of mitochondrial stressors oligomycin, FCCP, and rotenone + antimycin A. Cells were treated for three days with either DMSO, 25 µM of ARN, 10 nM of IACS, and a combination. (**B**) The metabolic parameters basal respiration, ATP production, and maximum respiration were calculated based on the Seahorse XF Cell Mito Stress Test Report Generator. Means ± SEM values of three independent experiments with three replicates each are shown (*n* = 9). Statistical significance is indicated by * *p* < 0.05; *** *p* < 0.001; **** *p* < 0.0001. (**C**) PNT1A, LNCaP, C4-2, and PC-3 cells stained for anti-TOM20 (red), visualizing mitochondrial networks. Nuclei were stained with DAPI (blue). White boxed areas were enlarged in the upper right corner. Scale bar: 50 µM. (**D**) Quantification of TOM20 total cellular fluorescence intensity. For each cell line, *n* = 40–50 cells were quantified by ImageJ; * *p* < 0.05; *** *p* < 0.001; **** *p* < 0.0001.

**Figure 4 cancers-15-05612-f004:**
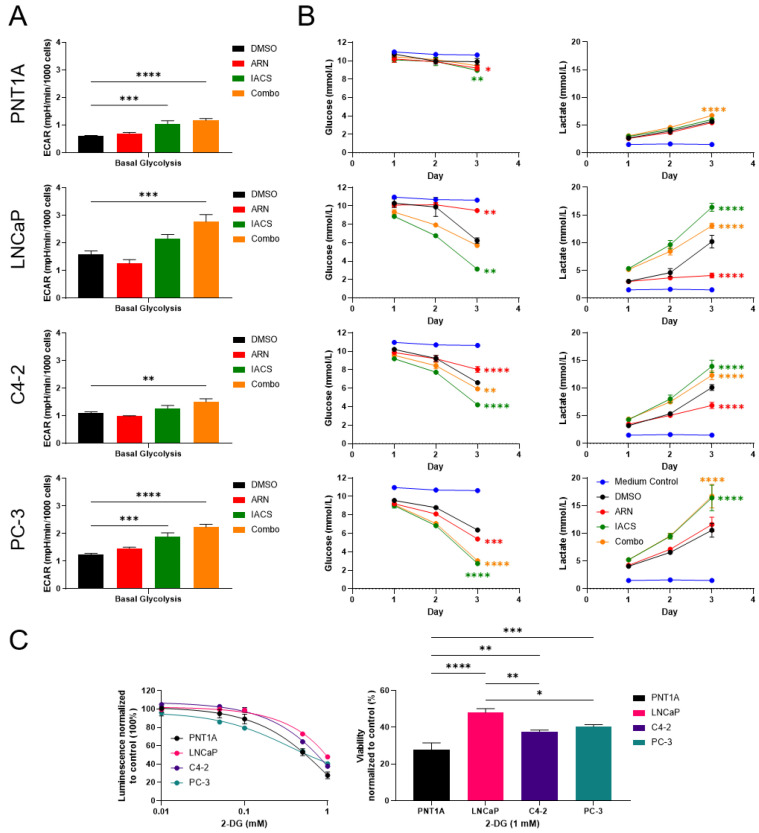
PCa cells have different specific glycolytic dependencies. (**A**) Extracellular acidification rate upon treatment with DMSO, ARN, IACS, and combination. Changes in basal respiration are presented in bar plots as means ± SEM (*n* = 9). (**B**) Extracellular medium analysis of glucose and lactate levels over a treatment period of three days measured by a Vi-CELL MetaFlex Bioanalyzer (*n* = 3). (**C**) A representative experiment is shown (*n* = 3). Medium analysis was performed twice independently. A 2-DG sensitivity test in PCa cell lines shows dose–response curves upon 2-DG addition (left). Cell proliferation normalized to untreated control is shown after three days of treatment with 1 mM of 2-DG (right). Data represent means ± SD. Statistical significance is indicated by * *p* < 0.05; ** *p* < 0.01; *** *p* < 0.001; **** *p* < 0.0001.

**Figure 5 cancers-15-05612-f005:**
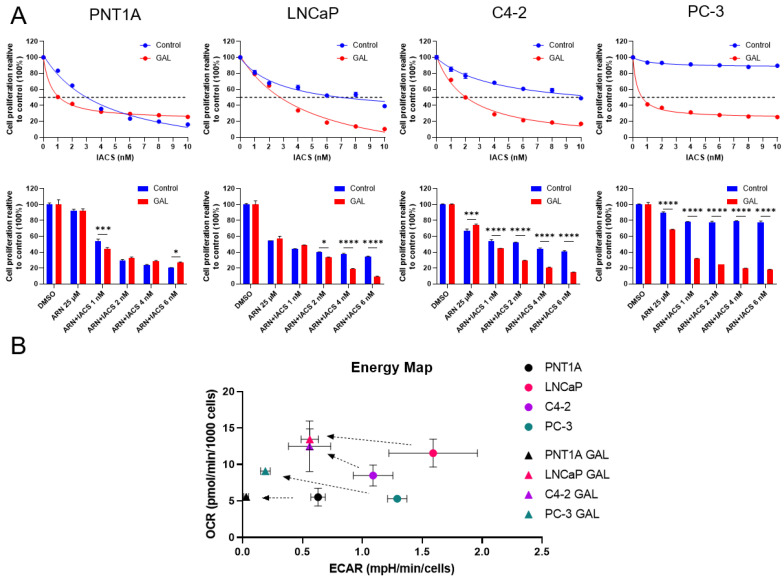
Galactose enhances treatment effect through metabolic rewiring. (**A**) Comparison of relative cell proliferation dose–response curves of increasing IACS concentration between control (glucose-supplemented) and GAL (galactose-supplemented) medium preparations (*n* = 3) (**top** row). Cell proliferation upon treatment with 25 µM of ARN and combinations with increasing concentrations of IACS for control and GAL (*n* = 3) (**bottom** row). Statistical significance is indicated by * *p* < 0.05; *** *p* < 0.001; **** *p* < 0.0001. (**B**) Energy map depicting ECAR values plotted against OCR of PCa cells grown in either normal or GAL-supplemented medium (*n* = 9). Arrows show a left shift in all cell lines towards decreased glycolysis and a slight upward shift in LNCaP, C4-2, and PC-3 cells indicating increased reliance of OXPHOS. Dots represent cells cultured in normal medium. Triangles mark cells cultured in glucose-free, galactose-supplemented medium.

## Data Availability

All relevant data are included within the manuscript.

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
