# Peer review of "Targeting Metabolic Vulnerabilities to Overcome Prostate Cancer Resistance: Dual Therapy with Apalutamide and Complex I Inhibition"

_cancers, 2023, doi:10.3390/cancers15235612_

Round 1

Reviewer 1 Report

Comments and Suggestions for Authors

This manuscript, “Targeting Metabolic Vulnerabilities to Overcome Prostate Cancer Resistance: Dual Therapy with Apalutamide and Complex I Inhibition”, focused on the metabolic behaviors of different types of prostate cancer cells and tested the potential synergistic effects of ARN and IACS. Overall, the study design was appropriate, and the findings provided useful insights.

The following minor comments or suggestions, if can be addressed, would further strengthen this manuscript.

1.     It would be helpful if the authors could report the sample size for each treatment group in all experiments.

2.     In Figures 3A and 3B, the scale of the Y-axis could be adjusted for better visualization.

3.     The quality of the figures could be improved.

Author Response

Thank you very much

Souzan

Reviewer 2 Report

Comments and Suggestions for Authors

In the present study, Baumgartner and colleagues analysed the therapeutic value of targeting metabolic vulnerabilities using a dual therapy with apalutamide, a complex I inhibition. The work is original, and the statistics and references are well-chosen. However, there are several major issues which need to be clarified before publication:

1.       In their first experiment, the group concludes that their experiments reveal that LNCaP cells have a specific fingerprint towards OXPHOS in an androgen-dependent fashion. They must be more careful with their conclusion based on the cell lines as they do not have an androgen-independent AR-positive cell line like C4-2.  

2.       The group uses the CellTiter-Glo 2.0 Assay for cell viability/proliferation experiments. This assay quantifies ATP to assess the cell viability. However, the group showed nicely that their treatments affect ATP production. Therefore, this assay may not be the correct assay to measure changes in cell viability. The proliferation assays should be repeated with cell counting or using incorporation assays. Alternatively, the group must show that the change in ATP does not interfere with the CellTiter-Glo 2.0 Assay and represents the change in proliferation.

3.       The group claims synergistic effects between Apalutamid and the mitochondrial complex inhibition exist. How did the group determine if these are synergistic or additive effects? Please use one of the literature-suggested models to prove drug synergism (e.g., PMID: 22737266).

4.       The group uses concentration, as they say, close to the IC50. Based on the literature, the IC50 for Apalutamide is between 10-200 nM (dependent on the chosen assay and cell system). Therefore, the group must show their IC50 calculations as they differ dramatically.

5.       Are the effects specific for apalutamide? Moreover, can androgen deprivation alone cause the same effects?

Author Response

Thank you very much

Souzan

Reviewer 3 Report

Comments and Suggestions for Authors

The work by Baumgartner and colleagues analyzes metabolic behavior (glycolysis/OXPHOS) and growth of “normal” prostate and prostate cancer cell lines in the absence or presence of inhibitors for the androgen receptor (AR) or the mitochondrial complex I (or a combination of both).  They propose that a combination of AR and OXPHOS inhibitors has the potential to restrict drug resistance and hinder progression to metastatic disease.

While targeting metabolic vulnerabilities is an exciting and very promising area of study in prostate and other cancers, overall this study is descriptive, confirmatory of established data in prostate cancer, and more importantly lacks mechanistic depth (i.e. relationship between AR signaling and metabolic adaptations) that could provide real avenues for novel prostate cancer treatment development.

1.   Figure 1: Malignant transformation in prostate involves conversion from glycolysis (normal) to oxidative phosphorylation (OXPHOS; early stages). Transformation to advanced prostate cancer promotes glycolysis, therefore different energy sources (OXPHOS, glycolysis, lipogenic..) are available at this stage. LNCaP cells are cancer cells and therefore have more OXPHOS as it has been widely shown before.  While it is appropriate that the authors confirm this, it just provides supplementary data.

2.  Figure 3 and 4:  the results with the OXPHOS inhibitor (IACS) and the combination therapy (ARN+IACS) do not seem to be significantly different, what would then be the point of using the combination therapy? In figure 4A for LNCaP, ECAR may be minimally higher (significant?) in cells treated with the combo vs. cells treated with IACS only, but the line graphs in figure 4B indicate that IACS treatment leads to higher glucose consumption and lactate production as compared to the combo treatment.

3.    Figure 2: Cell growth inhibition with the combination therapy is not very impressive. In fact, for the LNCaP and C4-2 cell lines, since the “ARN only” has an effect on cell growth, the synergistic effect in the graphs shown in figure 2D is not clear at all. These graphs are missing the ARN only curves and the statistics.

Moreover, Inhibitors of mitochondrial complex I have been previously shown to inhibit proliferation of prostate cancer cell lines and, as expected, many other cancers. In addition, toxicity of the IACS is a concern.

4.   The AR regulates metabolism, including glycolysis and OXPHOS, and therefore metabolic alterations are expected following AR inhibition (ARN) and reactivation (independent of mechanism) during progression to castration resistant prostate cancer. The choice of cell lines for this study is not very appropriate. If anything, a LNCaP-LNCaP-alb (or LNCaP95) pair will be easier to analyze since LNCaP-abl (or LNCaP95) derive from and are considered the CRPC form of LNCaP.

Overall this manuscript does not provide new concepts or mechanistic data on the relationship between AR signaling and metabolism that could offer potential novel opportunities for diagnosis and treatment.

Other comments:

Figure 1. The labels for B,C in the figure legend are switched.

Lines 208-210: “cells treated with ARN plus IACS showed significantly reduced cell 208 proliferation compared to ARN or IACS treated cells”. Figure 4A: There is no stats for the comparisons between IACS or the ARN and the combo and visually there is no significant difference between the IACS and the combo

Figure legend 4: please review.

Author Response

Thank you very much

Souzan

Reviewer 4 Report

Comments and Suggestions for Authors

Baumgartner et al. investigated metabolic reprogramming induced by mitochondrial and androgen receptor (AR) pathway inhibition in prostate cancer cells. They examined the effect of AR inhibition (ARN) and mitochondrial complex I inhibition (IACS) on several prostate cancer (PC) cells. They found that mitochondrial activity is upregulated in AR positive PC cells. ARN and IACS induced inhibition of AR-positive PC cells compared with AR negative cells. Cellular bioenergetics analysis revealed that IACS significantly repressed mitochondrial activity and enhanced glycolysis. In addition, they demonstrated that galactose supplementation disrupted compensatory glycolytic mechanism. Overall, they concluded that combinational inhibition of mitochondria and AR would be promising to overcome treatment resistance in PC.

This article is very intriguing by showing the subtype specific metabolic profiles of PC cells and new strategy to target PC cells. However, I think several points are not clearly described in this manuscript.

1)    (Figure 1D) This data is puzzling. Is this a gel image of western blot analysis? No explanation in figure legends. In addition, the number of replicates to make the graph is not described.

2)    (Figure 5B) The authors should add explanation what the round and triangle marks indicate. In addition, N number is also lacking.

3)    Using other AR negative cells such as DU145 and AR positive cells such as VCaP cells would be preferable.

4)    I wonder whether the authors examined the effect of androgen treatment on PC metabolic profile by seahorse analysis.

Author Response

Thank you very much

Souzan

Round 2

Reviewer 2 Report

Comments and Suggestions for Authors

This reviewer appreciates the answers to the mentioned issues. However, all of this additional information should be included in the manuscript (e.g., comparison of cell titer glow to cell counting, synergistic analysis and its interpretation, and discussion about effects of other AR inhibitors or ADT simulation). 

Author Response

Dear Reviewer,

Thank you very much for your input. Please see the attachment.

Best

Souzan

Reviewer 3 Report

Comments and Suggestions for Authors

No additional comments.

Author Response

Thank you very much.

Souzan

Round 3

Reviewer 2 Report

Comments and Suggestions for Authors

 Accept in present form

Author Response

Thank you very much.

Best

Souzan Salemi